# Antenatal screening of depressive and manic symptoms in south Brazilian childbearing women: A transversal study in advance of the pandemic scenario

Fernanda Schier de Fraga[1]◉*, Beatriz Souza Lima Wan-Dall[2]◉, Gabriel Henrique de Oliveira Garcia[2]‡, Henrique Pandolfo[2]‡, Adelyne Mayara Tavares da Silva Sequinel[3]◉, Pedro Alvin[2]‡, Eduardo Jonson Serman[1]◉, Vivian Ferreira do Amaral[1]◉

1 Department of Obstetrics and Gynaecology, Health Science Sector of Federal University of Parana, Curitiba, Parana, Brazil, 2 Department of Psychiatry, Health Science Sector of Federal University of Parana, Curitiba, Parana, Brazil, 3 Health Science Sector of Federal University of Parana, Curitiba, Parana, Brazil

◉ These authors contributed equally to this work.
‡ GHOG, HP, and PA also contributed equally to this work.
* feschier@hotmail.com

**Data Availability Statement:** Data cannot be shared publicly because of Research Ethics

## Abstract

### Background

The diagnosis of mood disorders (MD) during pregnancy is challenging and may bring negative consequences to the maternal-fetal binomial. The long waitlist for specialized psychiatric evaluation in Brazil contributes to the treatment omission. Almost 20.0% of women treated with antidepressants have a positive screening for bipolar disorder. Therefore, it has been recommended the investigation of depressive and bipolar disorder during prenatal care. Unfortunately, the screening for mood disorders is not a reality in Brazil and many childbearing women remain undiagnosed. The objective of this study is to observe the frequency of MD and the effectiveness of screening scales for routine use by health professionals during prenatal care in high-risk pregnancies.

### Methodology/Principal findings

This cross-sectional study included 61 childbearing women in their second trimester who were interviewed using the Edinburgh Postnatal Depression Scale (EPDS) and the Mood Disorder Questionnaire (MDQ). The cut-off point was EPDS ≥ 13 and MDQ ≥ 7 and the SCID-5 was the gold standard diagnosis. MD were diagnosed in 24.6% of the high-risk pregnancies. EDPS was positive in 19.7% and the frequency of major depression was 8.2%. 16.4% of the childbearing women were diagnosed with bipolar disorder, while MDQ was positive in 36.1%. 11.5% of the women had EPDS and MDQ positive. EPDS sensitivity was 80.0% and specificity 92.1%, whereas MDQ presented a sensitivity of 70.0% and specificity of 70.6%.

Committee of the University of Paraná Hospital ethical confidentiality. For researchers who meet the criteria for access to confidential data, it can be available from the Research Ethics Committee of the University of Paraná Hospital by the email cometica.saude@ufpr.br. The data underlying the results presented in the study are available for access in the GRADUATE PROGRAM OF TOCOGYNECOLOGY AND WOMEN'S HEALTH of Federal University of Parana Clinics Hospital, that can be contacted by the email pgtoco@ufpr.br.

**Funding:** The author(s) received no specific funding for this work.

**Competing interests:** The authors have declared that no competing interests exist.

## Conclusion/Significance

There is a high prevalence of MD in high-risk pregnancies. The routine use of EPDS simultaneously to MDQ during antenatal care is effective and plays an important role in early diagnosis, counselling, and promotion of perinatal mental health.

## Introduction

The gestational period increases the vulnerability to mood changes and may trigger signs and symptoms of infirmities that are often pre-existing, but never diagnosed [1]. Major depressive and bipolar disorder are important psychiatric conditions that can be identified in this period, affecting respectively 20.0% and 9.0–20.0% of childbearing women [2, 3]. Perinatal depression is manifested as a major depressive episode during pregnancy or until the first month of puerperium [4, 5]. Type I bipolar disorder is defined as at least a manic episode in life, interspersed with a history of hypomanic or major depressive episodes, whereas type II bipolar disorder predominates hypomanic and depressive episodes [2].

When not treated, there may be repercussions of mood disorder (MD) for children, such as low birth weight, prematurity, neuro-psycho-motor development deficit, difficulties in breastfeeding, and obesity during childhood [1, 6–9]. Mothers can be emotionally distanced from their children, presenting a high risk of suicide and infanticide [2]. Besides, the long waitlist for specialized psychiatric evaluation in Brazil leads to underdiagnosis and undertreatment.

The intensified contact with the health system during prenatal routine presents an opportunity for MD screening, as many national and international guidelines have been recommending [10–13]. In this context, screening scales such as the Edinburgh Postnatal Depression Scale (EPDS) and the Mood Disorder Questionnaire (MDQ) are useful tools in mental health care.

The EPDS, developed by Cox *et al.* in 1987 to help primary care providers detect symptoms of postpartum depression, was validated in Brazil in 2004 and became a meaningful method for screening for depressive symptoms in pregnancy and postpartum [14, 15]. The MDQ is a screening tool to investigate history of mania or hypomania [16], developed by Hirschfeld *et al.* in 2000, validated in Brazil in 2010 by Castelo *et al.*, and evaluated for application in the peripartum period by Frey in 2012 [6, 17, 18].

When investigating depressive symptoms during pregnancy, there is a high risk of interpreting a depressive phase of bipolar disorder as a major depressive disorder. Therefore, it supports simultaneous screening with the application of EPDS and MDQ during pregnancy [3].

More than half of women with postpartum depression report have experienced depressive episodes before or during pregnancy [1]. However, screening MD as part of the prenatal routine by health professionals is not a reality in Brazil [10–12].

Pointing out the importance of addressing this topic in obstetrics and the lack of Brazilian studies, this research aimed to verify the frequency of MD, evaluating EPDS and MDQ effectiveness for the screening and differential diagnosis of depressive symptoms in high-risk pregnant women of south Brazil.

## Materials and methods

### Participants, ethics and design

This cross-sectional study included 61 childbearing women, all over 18 years, with gestational age between 18 and 24 weeks who attended prenatal consultations at a public university

hospital reference for high-risk pregnancies in southern Brazil. These patients are referred by a Health Unit to the tertiary attention because they have or acquires a condition during pregnancy that puts them at higher risk for adverse events and requires more intensive care by obstetricians and other specialists [19]. The health care team must be prepared for many factors that may negatively impact high-risk pregnancies, whether they are clinical, obstetric, socioeconomic, or emotional [19]. According to Tsakiridis *et al.*, prenatal depression is more likely in high-risk pregnancies because they have more risk factors, suggesting greater attention to screening these patients [20].

This range of gestational age had the purpose of homogenize the sample, which was collected from August 2018 to August 2019. Women who were on drug treatment for any psychiatric disorder at the moment of the invitation were excluded from the study to avoid bias during the application of the screening scales.

Considering the outlines of the analytical cohort research and diagnostic test, two hypotheses were built. The first is related to the cohort study, considering that the frequency of MD during pregnancy is high, and clinical and epidemiological factors may be associated with its development. The second hypothesis refers to the diagnostic test study, considering effective EPDS and MDQ for the screening of depressive and manic symptoms in this population when using a structured clinical interview as the gold standard.

This study was approved by the Institution's Research Ethics Committee in June of 2018 with approval number 61858716.8.0000.0096.

## Clinical assessments

After signing the consent form and receive a copy, the questionnaires EPDS and MDQ were applied to the childbearing women. If she didn't accept to participate, she continued the prenatal follow-up as habitual.

The EPDS was validated for the Portuguese language by Santos *et al.* and it consists of 10 questions about how the person feels about depressive symptoms in the last week [15]. The scoring is performed according to the responses obtained, making up a maximum of 30 points. A score of 13 points or more was considered positive for depressive symptoms in this study since it indicates a high probability of major depressive disorder [14, 15]. The EPDS scale was the instrument of choice due to its high levels of sensitivity and specificity found in the literature, and its well-established use during pregnancy and puerperium in Brazil.

The MDQ consists of a questionnaire of three parts: the first one has 13 questions with affirmative or negative answers about the history of mania and hypomania symptoms, and the two other parts investigate the frequency and consequences of these symptoms in women's life. Frey *et al.* identified that applying only the first part of the MDQ makes it more sensitive than the complete questionnaire, which has been used in this study as a positive screening for a history of mania if the women scored seven or more affirmative answers [18]. This scale was chosen because international studies are validating its use in the perinatal period, being the most used scale for the investigation of manic symptoms. The absence of Brazilian studies using this scale in childbearing women made its validation in high-risk pregnancies possible.

After the interview with the first applicator, regardless of the screening result, all women were referred for evaluation by a resident physician in psychiatry, who conducted the structured interview (SCID-5), in another outpatient room, without knowing the results of the screening.

The SCID-5® clinical version was used as the gold standard for MD diagnostic in childbearing women, promoting uniformity in the selected diagnosis [21]. In this study, only the mood disorders module was applied. Referrals for a specialized follow-up were performed in all MD diagnoses.

### Data analysis

The indexes of accuracy, sensitivity, specificity, positive predictive value, negative predictive value, false positive, and false negative were estimated considering SCID-5® as the gold standard.

Measures of central tendency and dispersion were expressed as means and standard deviation (mean ± SD) for continuous variables with symmetrical distribution and as medians, interquartile range (IQR) for those with the asymmetric distribution. Categorical variables were expressed as absolute and relative frequency. Univariate logistic regression was applied to estimate the probability of positive screening for a current depressive episode (SCID-5) according to EPDS.

The sample size was calculated to evaluate the accuracy of the screening scales for MD diagnosis, with an estimated sensitivity of 90.0% and the suggested sample size was 60 subjects. All statistics were two-sided, and a 0.05 significance level was used (*Statistica* 10.0—*Statsoft*®).

## Results

The sample consisted mostly of multigravidas (98.0%)–Table 1. Considering that 63.8% of the patients did not use contraception, about 37.5% had unplanned pregnancies. The most important risk factors for pregnancy were hypertension (22.9%), previous preterm labor (19.7%), complications at last delivery (18.0%), hypothyroidism (16.4%), and obesity (13.1%).

Psychiatric disorders in the family were found in 32 subjects (52.4%). In 26 subjects, the informant indicated the family member, mostly the mother (43.2%) and uncles or aunts (30.8%). 21.3% had used illegal substances, 14.7% had attempted suicide, and 14.7% suffered from anxiety.

The frequency of a depressive episode by SCID-5 was 16.4%, with 8.2% of the sample diagnosed with a current isolated depressive episode and 8.2% with a depressive episode of bipolar disorder. The frequency of bipolar disorder was 16.4% (Table 2).

Five childbearing women answered "yes" to EPDS question 10, referring to the idea of harming themselves—one responded she had frequent thoughts in the past seven days and the others have had very few times these thoughts.

When evaluating the best cut-off point of the EPDS for positive screening, it was observed that, with a score of 10, the probability of estimated MD was about 15.0%, increasing progressively from this score (Fig 1).

The EPDS score varied from 0 to 24 points, with positive screening in 19.7% of the childbearing women. The MDQ score was between 0 to 12, with positive screening in 36.1% (Table 1). Positive screening by both scales was present in 7 subjects (11.5%).

Considering a score greater or equal to 13 and using SCID-5 as the gold standard, EPDS showed an accuracy of 90.2% for the identification of a depressive episode, with a sensitivity of 80.0%, specificity of 92.1%, false-positive of 33,3%, and false-negative of 4.1% (Table 2). With the cut-off point > 10, the accuracy was 85.2%, the sensitivity was 90.0%, the specificity was 84.3% and the false-positive index was 47.0%. With a cut-off point > 14 these values were 93.3%, 77.7%, 96.1% and 22.2%, respectively. The cut-off point ≥ 7 of the MDQ showed an accuracy of 70.5% for identifying bipolar disorder, with a sensitivity of 70.0%, a specificity of 70.6%, a false-positive of 68.2%, and a false-negative of 7.7% (Table 3).

## Discussion

High-risk pregnant women are more likely to present depressive symptoms. The review by Tsakiridis *et al.* found a prevalence from 12.5 to 44.2% of depression in childbearing women who had at least one risk factor during prenatal care [20]. Considering the presence of a

**Table 1. Sample features.**

| Features | Average ± SD /n (%) /median (IQR) |
|---|---|
| Age (years) | 29,9 ± 8,2 |
| Race | |
| White | 38 (62.4%) |
| Yellow | 1 (1.6%) |
| Brown | 18 (29.5%) |
| Black | 4 (6.5%) |
| Occupation | |
| Unemployed | 9 (14.7%) |
| Working | 35 (57.4%) |
| Student | 2 (3.3%) |
| Housewife | 15 (24.6%) |
| Scholarity | |
| 1st degree | 24 (39.3%) |
| 2nf degree | 25 (41.0%) |
| Superior | 12 (19.7%) |
| Marital status | |
| Single without permanente partner | 8 (13.1%) |
| Single with permanent partner | 6 (9.8%) |
| Domestic partnership | 23 (37.7%) |
| Married | 24 (39.4%) |
| Number of people in the residence | 2 (2–4) |
| Monthly family income (minimum salary in Brazil is approximately $200 dollars) | |
| Up to 1 minimum salary | 9 (14.7%) |
| 1 to 3 | 35 (57.4%) |
| 3 to 6 | 16 (26.2%) |
| 6 to 10 | 1 (1.6%) |
| Works Away from home | 25 (41.0%) |
| Stopped working due to pregnancy | 12 (19.7%) |

current depressive episode, whether due to major depression or depressive episode of bipolar disorder, we observed a 16.4% prevalence in our study, a value similar to that described by Gavin et al., who suggested that 18.4% of women are affected by depressive episodes during pregnancy [22].

In this sample, 8.2% of high-risk pregnant women were diagnosed with an isolated depressive episode by SCID-5, similar to the prevalence observed in the second trimester of pregnancy by Gavin et al. (8.5%) and Usuda et al. (9.5%) [22, 23]. These results are inferior to those of major depression in the second trimester obtained in Brazilian childbearing women after application of the MINI-Plus Interview by Castro and Couto et al. and Brancaglion et al., that achieved prevalence of 17.3% and 21.7%, respectively [4, 5].

The frequency of depression within the trimesters of pregnancy is also demonstrated in other studies. Gavin et al. found 11.0% of major depression in the first trimester and 8.5% in the second trimester, diagnosed through structured clinical interviews [22]. Bennett et al., demonstrated a prevalence of 7.4% in the first trimester, followed by 12.8% in the second trimester and 12.0% in the third trimester [24]. Other studies show the prevalence of antepartum

**Table 2. Scores and results of the Edinburgh Postpartum Depression Scale, Mental Disorder Questionnaire and frequency of depressive disorder and bipolar disorder.**

| Screening and Scores | n (%)/median (IQR) |
|---|---|
| SCID-5 | |
| Current depressive episode | 10 (16.4%) |
| Bipolar affective disorder | 10 (16.4%) |
| Isolated current depressive episode | 5 (8.2%) |
| Isolated bipolar affective disorder | 5 (8.2%) |
| Current depressive episode and bipolar affective disorder | 5 (8.2%) |
| Current depressive episode or bipolar affective disorder | 15 (24.6%) |
| EPDS screening | 5 (1–11) |
| Positive | 12 (19.7%) |
| Negative | 49 (80.3%) |
| MDQ screening | 5 (2–7) |
| Positive | 22 (36.1%) |
| Negative | 39 (63.9%) |
| Positive isolated EPDS | 5 (8.2%) |
| Positive Isolated MDQ | 15 (24.6%) |
| Positive EPDS and MDQ | 7 (11.5%) |
| Positive EPDS or MDQ | 27 (44.3%) |

depressive episodes in low-income countries of 25.8%, while in the postpartum period it decreases to 19.7% [1].

Almost 60.0% of postpartum depressive episodes began in the pregnancy, which reinforces the importance of investigating these symptoms during prenatal follow-up since the diagnosis of MD in this period has been associated with maternal and fetal complications [1, 6–9].

The highest prevalence of mood disorders in childbearing women concerning menacme may occur because during the gestational period the limits between the physiological and pathological aspects of mental health are narrow care [1].

The prevalence of bipolar disorder in the general population varies from 1.0–2.0%, whereas in the United States it reaches 4.4%, with a similar incidence between genders [8, 9]. However, depressive episodes, precipitous mood swings, and mania or hypomania are more frequent in women [9]. Considering that pregnancy is a time of vulnerability to relapse due to hormonal and circadian rhythm changes, the prevalence of the bipolar disorder in the perinatal period varies between 2.0–8.0% [7, 8].

It is believed that more than half of women who present postpartum depressive symptoms suffer from bipolar disorder, and about 60.0 to 70.0% of women diagnosed with bipolar disorder had presented episodes of mood changes during pregnancy and postpartum [3, 25]. Although our study evaluated women during pregnancy, we observed a similar situation, since 8.2% of 16.4% of childbearing women who had a depressive episode were diagnosed with bipolar disorder by SCID-5 and 8.2% were diagnosed with major depression (50.0%).

Given the direct contact with health professionals, the perinatal period is the most suitable for the investigation of mood disorders in childbearing women, and it has been stimulated by many guidelines.

The *United States Preventive Services Task Force* (USPSTF) recommends the screening for depression during pregnancy and postpartum, associated to counselling interventions [10]. The American College of Obstetricians and Gynaecologists (ACOG) recommends screening for depression at least once during the peripartum period, indicating that childbearing women

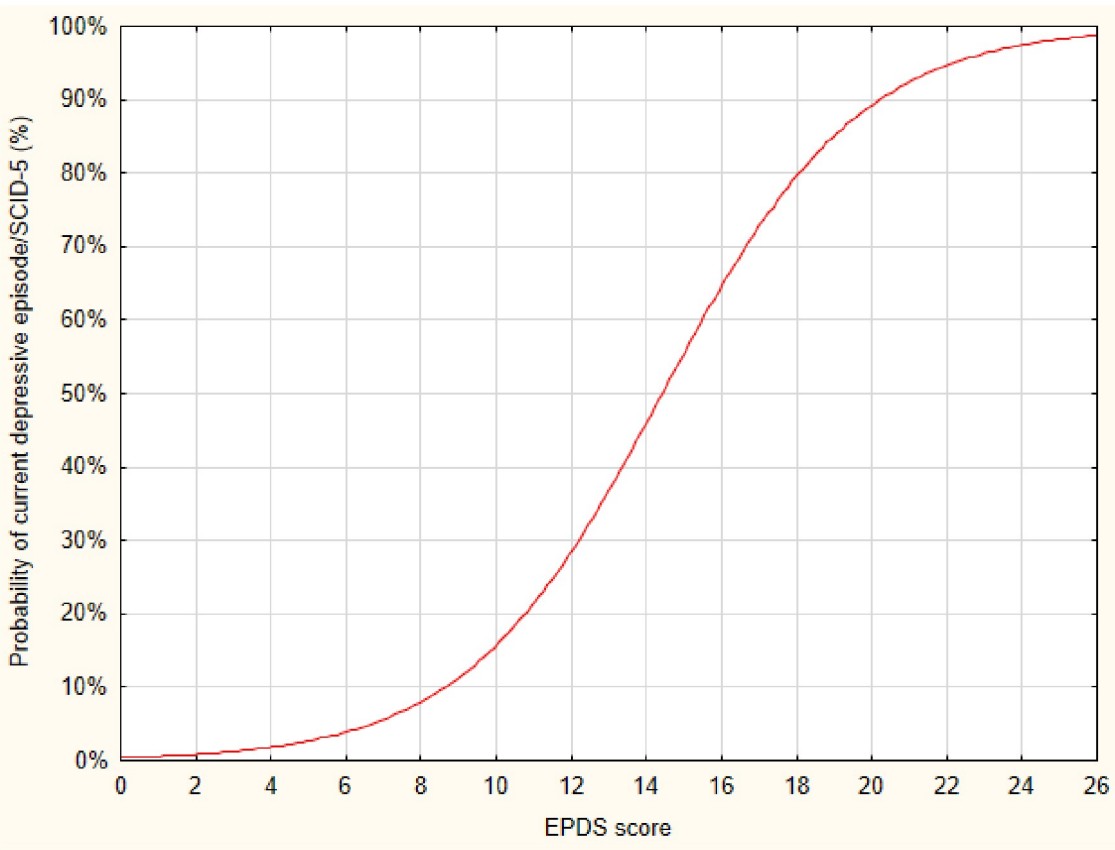

**Fig 1. Probability of positive screening for current depressive episode (SCID-5) according to EPDS.** Logistic regression: p < 0,001 dependent variable: depressive episode/SCID-5; independent variable: EPDS score.

with suspected mania should receive specialized care before starting treatment [13]. The Canadian Network for Mood and Anxiety Treatments (CANMAT) suggests the screening for bipolar disorder with MDQ in women with depressive symptoms [12].

In Brazil, the Society of Obstetrics and Gynaecology of São Paulo recommends screening for postpartum depression, without mentioning the gestational period, advising the

**Table 3. Accuracy of Edinburgh Postpartum Depression Scales and Mental Disorder Questionnaire for the identification of depressive episode and bipolar disorder.**

| EPDS Scale | | Depressive Episode SCID-5 | | Total |
|---|---|---|---|---|
| | | Positive | Negative | |
| EPDS | Positive | 8 (66.7%) | 4 (33.3%) | 12 (19.7%) |
| | Negative | 2 (4.1%) | 47 (95.9%) | 49 (80.3%) |
| Total | | 10 (16.4%) | 51 (83.6%) | 61 (100.0%) |
| MDQ Scale | | Bipolar Disorder SCID-5 | | Total |
| | | Positive | Negative | |
| MDQ | Positive | 7 (31.8%) | 15 (68.2%) | 22 (36.1%) |
| | Negative | 3 (7.7%) | 36 (92.3%) | 39 (63.9%) |
| Total | | 10 (16.4%) | 51 (83.6%) | 61 (100.0%) |

obstetricians to be aware of the risk factors and refer the woman to mental healthcare professionals when needed [11].

Mental health services are usually restricted to the treatment of psychiatric disorders already installed or of greater severity. Thereby, Santos *et al.* reported great benefits of EPDS for depression screening since it can be executed by any health professional [15].

In our study, EPDS was positive in 19.7% of childbearing women, similar to the studies of Da Silva *et al.*, Melo *et al.*, and Silva *et al.*, who obtained rates of 17.9%, 24.3%, and 20.5% respectively [26–28]. The high sensitivity and specificity of EPDS, such as observed by other authors (Table 4), suggests that even though the scale was applied to a high-risk pregnant women population, it had not overestimated the results.

Despite some authors point to a score of 10 as the cut-off point of EPDS, in our study it had a low probability of success (about 15.0%), justified by the less accuracy, less specificity, a higher rate of false positives, but greater sensitivity. To encourage the screening for MD, it must be considered that the tests of high sensitivity, but low specificity, can lead to high rates of false positives, conducting to a greater number of subjects for evaluation by specialists, unduly overburdening the health system, generating emotional damage and higher costs. The cut-off points of 14 increases accuracy, specificity and further decreases the false positive (22.2%) but greatly decreases sensitivity (77.7%). Thus, the cut-off points of 13 seems more effective, since it increases accuracy, specificity, and sensitivity, with a more acceptable false-positive rate.

Compared to the general female population, childbearing women have a lower rate of suicide; however, suicidal ideation may be more representative among the ones with depressive disorder [1, 25]. Nevertheless, suicide is a leading cause of direct maternal death in the first year of postpartum in several countries [29–32]. Therefore, it is essential to ask about suicidal thoughts in patients with depressive symptoms. In our study, five women responded positively to the tenth question of EPDS, pointing out that any affirmative answer to this question presents a high risk of suicidal ideation. Among the five (8.2%), three received a diagnosis of bipolar disorder and one of depressive disorder according to the criteria proposed by DSM-5 in SCID-5. All were referred for psychiatric follow-up. In a study during the second trimester of pregnancy, 6% of the sample of Castro e Couto *et al.* answered "yes" to the tenth EPDS question, which correlated strongly with the diagnosis of MD [33].

Frey *et al.* found in a population of childbearing women and women who had recently given birth that the cut-off $\geq$ 7 had 89.0% sensitivity, 84.0% specificity, and 43.0% positive predictive value [18]. In our study, MDQ performed a sensitivity of 70.0%, specificity of 70.6%, false positive of 68.2%, and false-negative of 7.7%. The validation of MDQ in a general psychiatric Brazilian population presented a sensitivity of 91.0% and specificity of 70.0%, using cut-off $\geq$ 7 associated with the two supplementary questions [6]. There are no Brazilian studies in the literature using MDQ in childbearing women. Our research is pioneer in demonstrating

**Table 4. EPDS sensibility and specificity in the literature.**

| Scientific papers | Year | Country | Cut-off | Sensibility | Specificity |
|---|---|---|---|---|---|
| Usuda *et al.* [23] | 2017 | Japan | 13 | 90.0% | 79.0% |
| Castro e Couto *et al.* [4] | 2015 | Brazil | 11 | 81.0% | 73.0% |
| Brancaglion *et al.* [35] | 2013 | Brazil | 9 | 80.0% | 70.0% |
| Silva *et al.* [28] | 2012 | Brazil | 13 | 59.5% | 88.4% |
| Melo *et al.* [27] | 2012 | Brazil | 13 | 75.0% | 81.0% |
| Felice *et al.* [36] | 2006 | Malta | 13 | 75.0% | 95.8% |
| Da Silva *et al.* [26] | 1998 | Brazil | 13 | 73% | 90.5% |

that this tool is useful for screening of manic episodes by health professionals. It might not only help in the differential diagnosis of MD, but might encourage the referring of childbearing women with score ≥7 for specialized clinical evaluation.

In a prospective study, Masters *et al.* observed that 18.8% of childbearing and postpartum women had MDQ ≥ 7, half of which we had found in our sample, in which 36.1% screened positive with MDQ (≥ 7). The diagnosis of bipolar disorder by SCID-5 occurred in 16.4% of our sample, higher than the prevalence described in other studies [7].

The problem of the misdiagnosis of bipolar disorder is the erroneous treatment. Studies evidence that almost 20.0% of women treated with antidepressants have a positive screening for bipolar disorder [24]. Therefore, it is fundamental to have good accuracy for differential diagnosis in screening tools. The accuracy of the MDQ in this study was 70.5%, which means that almost 30.0% of women may not have the correct diagnosis. The complementary psychiatric evaluation for those who score positive for EPDS but negative for MDQ is a solution for fewer false negatives and is recommended by many guidelines [10].

According to Merril *et al.*, if only women with depressive symptoms were screened for manic episodes, approximately one-third of bipolar disorder subjects would be missed [24]. In this study, 24.6% of childbearing women would not be identified as a possible diagnosis of bipolar disorder if only depressive symptoms were screened, using EPDS. Merril *et al.* observed that 21.4–57.1% of childbearing women could have this diagnostic omission [24].

The individual use of the EPDS can cause a misdiagnosis, as depressive symptoms in pregnancy may be associated with episodes of hypomania [34]. Since MDQ can explore the past psychiatric history, it adds up to the screening and management of the patients. Guidelines recommend depressive and bipolar disorder investigation during prenatal care, which encourages the screening of depressive and manic symptoms during antenatal consults using EPDS and MDQ [7, 9, 34].

The obstetricians' and midwives' role in perinatal mental health should be reinforced by MD screening as an obligatory intervention during prenatal care, thereby reducing morbidity and improving the quality of life of childbearing women and their families. With the possible increase in referrals for follow-up with mental health specialists, it is necessary to train health professionals to carry out the first approach, speeding up diagnosis and early implementation of therapy, since women in the perinatal period have continuous contact with the health system.

The limitation of this study is that we cannot extrapolate the validity of the screening using MDQ and EPDS regarding all pregnant Brazilian women. However, it points out its accuracy in high-risk pregnancies during the second trimester. Furthermore, given the prevalence of the bipolar disorder in this study, even eight times greater than in the general population, the MDQ results should be viewed with caution as a result of the sample size, although it was sufficient to indicate the significant contribution of the EPDS instrument to the diagnosis. Future studies using these screening tools at routine second-trimester examinations could help expand the sample to include childbearing women at all levels of care. They could also highlight the public benefits of perinatal mental health and demonstrate that health professionals can use simple tools to promote mental wellbeing. Another relevant topic is that these MD screening scales evaluate psychiatric symptoms, nor the diagnosis of MD, which needs a longitudinal follow-up.

We can use our study to inspire health professionals to amplify their mindset about perinatal mental health. Although in a pre-pandemic scenario, mental health was already of paramount importance, now social isolation and other events such as trauma, worry, and grief as a result of the pandemic reinforce the need to encourage public policies worldwide on mental health education and the importance of patient-centered care.

## Conclusion

Mood Disorder was observed in about 25.0% of childbearing women and the simultaneous application of the EPDS and MDQ scales in high-risk pregnant Brazilian women proved to be adequate for the screening of depressive symptoms and manic history in the second trimester of pregnancy. These screening tools should be used routinely by health professionals, stimulating women-centred care and better maternal outcomes.

## Acknowledgments

We thank Dr. Dirceu Zorzetto Filho, Dr. Leandro Michelon and Dr Monica Lima for the useful suggestions and special thanks to the patients that kindly participated in this study.

## Author Contributions

**Conceptualization:** Fernanda Schier de Fraga, Eduardo Jonson Serman, Vivian Ferreira do Amaral.

**Data curation:** Fernanda Schier de Fraga, Beatriz Souza Lima Wan-Dall, Gabriel Henrique de Oliveira Garcia, Henrique Pandolfo, Adelyne Mayara Tavares da Silva Sequinel, Pedro Alvin, Eduardo Jonson Serman, Vivian Ferreira do Amaral.

**Formal analysis:** Fernanda Schier de Fraga, Beatriz Souza Lima Wan-Dall, Adelyne Mayara Tavares da Silva Sequinel, Eduardo Jonson Serman, Vivian Ferreira do Amaral.

**Investigation:** Fernanda Schier de Fraga, Eduardo Jonson Serman, Vivian Ferreira do Amaral.

**Methodology:** Fernanda Schier de Fraga, Beatriz Souza Lima Wan-Dall, Gabriel Henrique de Oliveira Garcia, Henrique Pandolfo, Adelyne Mayara Tavares da Silva Sequinel, Pedro Alvin, Eduardo Jonson Serman, Vivian Ferreira do Amaral.

**Project administration:** Fernanda Schier de Fraga, Eduardo Jonson Serman, Vivian Ferreira do Amaral.

**Resources:** Fernanda Schier de Fraga, Eduardo Jonson Serman, Vivian Ferreira do Amaral.

**Software:** Fernanda Schier de Fraga, Beatriz Souza Lima Wan-Dall, Adelyne Mayara Tavares da Silva Sequinel, Eduardo Jonson Serman, Vivian Ferreira do Amaral.

**Supervision:** Fernanda Schier de Fraga, Eduardo Jonson Serman, Vivian Ferreira do Amaral.

**Validation:** Fernanda Schier de Fraga, Gabriel Henrique de Oliveira Garcia, Henrique Pandolfo, Pedro Alvin, Eduardo Jonson Serman, Vivian Ferreira do Amaral.

**Visualization:** Fernanda Schier de Fraga, Eduardo Jonson Serman, Vivian Ferreira do Amaral.

**Writing – original draft:** Fernanda Schier de Fraga, Beatriz Souza Lima Wan-Dall, Gabriel Henrique de Oliveira Garcia, Henrique Pandolfo, Adelyne Mayara Tavares da Silva Sequinel, Pedro Alvin, Eduardo Jonson Serman, Vivian Ferreira do Amaral.

**Writing – review & editing:** Fernanda Schier de Fraga, Eduardo Jonson Serman, Vivian Ferreira do Amaral.

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
