## [Decision Letter · Decision Letter 0]

20 Sep 2021

PONE-D-21-26356Antenatal screening of depressive and manic symptoms in south Brazilian childbearing women: A transversal study in advance of the pandemic scenarioPLOS ONE

Dear Dr. Schier de Fraga,

Thank you for submitting your manuscript to PLOS ONE. After careful consideration, we feel that it has merit but does not fully meet PLOS ONE’s publication criteria as it currently stands. Therefore, we invite you to submit a revised version of the manuscript that addresses the points raised during the review process.

More specifically,  it would be interesting to better justify why your sample size is small and discuss this point if it could not be increased in a reasonable time. 

We look forward to receiving your revised manuscript.

Kind regards,

Raoul Belzeaux, M.D.

Academic Editor

PLOS ONE

Journal Requirements:

Reviewers' comments:

Reviewer's Responses to Questions

**Comments to the Author**

1. Is the manuscript technically sound, and do the data support the conclusions?

Reviewer #1: Partly

Reviewer #2: Yes

2. Has the statistical analysis been performed appropriately and rigorously? 

Reviewer #1: No

Reviewer #2: I Don't Know

3. Have the authors made all data underlying the findings in their manuscript fully available?

Reviewer #1: Yes

Reviewer #2: Yes

4. Is the manuscript presented in an intelligible fashion and written in standard English?

Reviewer #1: Yes

Reviewer #2: Yes

5. Review Comments to the Author

Reviewer #1: The manuscript entitled ‘Antenatal screening of depressive and manic symptoms in south Brazilian childbearing women: A transversal study in advance of the pandemic scenario’ with the aim to observe the frequency of Mood Disorder and the effectiveness of screening scales for routine use by health professionals during prenatal care in high-risk pregnancies.

The manuscript can be improved based on the comments below.

Abstract

Methodology/Principal Findings

The sentence ‘This is a transversal study including 61 childbearing women in the second trimester that were submitted to the Edinburgh Postnatal Depression Scale (EPDS) and Mood Disorder Questionnaire (MDQ)’ requires revision.

Introduction

Line 69, the word validated in Brazil to be added.

Materials and methods

Sample size

Line 102, more information to be provided e.g 1 or 2 tailed test. The sample size figures derived to be stated.

Line 103, for the margin of error, (confidence interval) to be removed.

Line 105, reference number to be included.

Data analysis

Line 147, Table 1, IQR could be used along with median.

Line 149-153, there are many statistical tests mentioned but not all were used in the analyses. For the statistical tests that were not used, it has to be removed. If the statistical test was used, it is to be clearly denoted in the results section and mention in the statistical analyses section.

Line 155, statistical software and the level of significance to be stated.

Results

Baseline characteristics of the subjects to be included and presented in table form.

Line 171, Graph 1 to be replaced with Figure 1.

Line 182, the cut off for Bipolar Disorder SCID-5 to be stated.

Table 2 Depressive Episode SCID-5, the percentage figures were presented based on row but for total, the percentage figures was based on column. This needs to be standardized.

Graph 1, the dependent and independent variable to be clearly denoted in the footnote.

Discussion

Line 255, the word 'chart 1' to be replaced with table and respective table number or to be omitted.

Line 259 Chart 1, typo Sensibility.

Decimal points for percentages to be standardized.

Not all references conform to PLoS One format.

Reviewer #2: Perinatal period is a time of great vulnerability for women, especially when they present a history of psychiatric disorder.

Mood disorder relapses are frequent in perinatal period. Howewer the screening for mood disorders has to be improved in gestational period to prevent several consequences for mothers and children and to differentiate bipolar disorder from other conditions.

This study has two objectives. The first is to estimate the frequency of mood disorder in perinatal period. The second is to evaluate EPDS and MDQ effectiveness in the screening of mood disorders in gestational period.

This study is very interesting and very clear. Howewer, I have some comments.

L60 "MDQ is a screening tool for episodes of mania" => The MDQ is a screening tool for history of mania or hypomania

L87 "This cross-sectional study included 61 childbearing women"

The sample size is small and should be discussed

L89 High risk pregnancies group should be defined. Moreover, the reason why high risk pregnancies have been chosen to evaluate MDQ and EPDS effectiveness should be explained.

L279 "Compared to the general female population, childbearing women have lower suicide rates."

Nevertheless, suicide is one of the leading cause of maternal death in several countries.

• Knight M, Bunch K, Tuffnell D, Shakespeare J, Kotnis R, Kenyon S, Kurinczuk JJ, on behalf of MBRRACE-UK. Saving Lives, Improving Mothers’ Care - Lessons learned to inform maternity care from the UK and Ireland Confidential, Enquiries into Maternal Deaths and Morbidity 2015-17, 2019

• SepVangen S, Bødker B, Ellingsen L, Saltvedt S, Gissler M, Geirsson RT, Nyfløt LT. Maternal deaths in the Nordic countries, Acta Obstet Gynecol Scand, 2017

• Amélie Boutin PhD , Arlin Cherian MPH , Jessica Liauw MD, MHSc ,Susie Dzakpasu PhD ,Heather Scott MD ,Michiel Van den Hof MD ,Jocelynn Cook PhD ,Jennifer Blake MD , K.S. Joseph MD, PhD , for the Canadian Perinatal Surveillance System(Public Health Agency of Canada), Database autopsy: An efficient and effective confidential en-quiry into maternal deaths in Canada, Journal of Obstetrics and Gynaecology Canada, 2020

• Deneux-Tharaux C, Morau E, Dreyfus M; pour le Cnemm. Mortalité maternelle en France 2013–2015 : un profil qui évolue [Maternal mortality in France 2013-2015: An evolving profile]. Gynecol Obstet Fertil Senol. 2021 Jan

6. PLOS authors have the option to publish the peer review history of their article (what does this mean?). If published, this will include your full peer review and any attached files.

Reviewer #1: No

Reviewer #2: **Yes: **Elsa MOREAU

---

## [Author Response · Author response to Decision Letter 0]

18 Oct 2021

Response to the reviewers

Dear Editor, 

Raoul Belzeaux 

Plos One

Thank you for the pertinent review and for the opportunity of improving our paper entitled “Antenatal screening of depressive and manic symptoms in south Brazilian childbearing women: A transversal study in advance of the pandemic scenario.”

We appreciated the comments and have accepted all the suggestions. 

 Please find below our answers and the list of changes point by point. These changes are marked in blue in the manuscript.

Reviewer#1

Remark1# Abstract - Methodology/Principal Findings

The sentence “This is a transversal study including 61 childbearing women in the second trimester that were submitted to the Edinburgh Postnatal Depression Scale (EPDS) and Mood Disorder Questionnaire (MDQ)” requires revision.

 Our answer: The sentence was corrected to: “This cross-sectional study included 61 childbearing women in their second trimester who were interviewed using the Edinburgh Postnatal Depression Scale (EPDS) and the Mood Disorder Questionnaire (MDQ).” (Line 29 to 31).

Remark 2# Introduction

Line 69, the word validated in Brazil to be added.

 Our answer: The suggested word was included and the sentence improved to: “The EPDS, developed by Cox et al. in 1987 to help primary care providers detect symptoms of postpartum depression, was validated in Brazil in 2004 and became a meaningful method for screening for depressive symptoms in pregnancy and postpartum.” (Line 63 to 66).

Remark 3# Materials and methods

# 3.1 Sample size

 Our answer: The sentence has been modified to better explain the calculation of the minimum sample size and the limitation for extending the sample due to the pandemic scenario: “The sample size was calculated to evaluate the accuracy of the screening scales for MD diagnosis, with an estimated sensitivity of 90.0% and the suggested sample size was 60 subjects. All statistics were two sided, and a 0.05 significance level was used (Statistica 10.0 - Statsoft®).” (Line 147 to 150).

Nevertheless, we attached possible ways for future studies with increased sample after the pandemic scenario that might be compared with this study and included a sentence in the limitation study: “Furthermore, given the prevalence of the bipolar disorder in this study, even eight times greater than in the general population, the MDQ results should be viewed with caution as a result of the sample size, although it was sufficient to indicate the significant contribution of the EPDS instrument to the diagnosis. Future studies with increased sample after the pandemic scenario using these screening tools at routine second-trimester examinations could help expand the sample to include childbearing women at all levels of care. They could also highlight the public benefits of perinatal mental health and demonstrate that health professionals can use simple tools to promote mental wellbeing”. (Line 320 to 327).

# 3.2 Line 102, more information to be provided e.g 1 or 2 tailed test. The sample size figures derived to be stated.

 Our answer: The necessary information has been added and the sentence moved to the data analysis session: “The sample size was calculated to evaluate the accuracy of the screening scales for MD diagnosis, with an estimated sensitivity of 90.0% and the suggested sample size was 60 subjects. All statistics were two-sided, and a 0.05 significance level was used (Statistica 10.0 - Statsoft®)” (Line 147 to 150).

# 3.3 Line 103, for the margin of error, (confidence interval) to be removed.

 Our answer: The sentence was removed from the manuscript.

# 3.4 Line 105, reference number to be included.

Our answer: The reference number was included: “This study was approved by the Institution's Research Ethics Committee in June of 2018 with approval number 61858716.8.0000.0096”. (Line 106).

# 3.5 Data analysis Line 147, Table 1, IQR could be used along with median.

 Our answer: The IQR was included in the sentence as well as in the tables 1 and 2: “Measures of central tendency and dispersion were expressed as means and standard deviation (mean±SD) for continuous variables with symmetrical distribution and as medians, interquartile range (IQR) for those with the asymmetric distribution. Categorical variables were expressed as absolute and relative frequency.” (Line 141 to 144).

# 3.6 Line 149-153, there are many statistical tests mentioned but not all were used in the analyses. For the statistical tests that were not used, it has to be removed. If the statistical test was used, it is to be clearly denoted in the results section and mention in the statistical analyses section.

 Our answer: The statistics tests were not used have been removed as well as clearly described in the data analysis session and in the figure 1 legend: “Measures of central tendency and dispersion were expressed as means and standard deviation (mean ± SD) for continuous variables with symmetrical distribution and as medians, interquartile range (IQR) for those with the asymmetric distribution. Categorical variables were expressed as absolute and relative frequency. Univariate logistic regression was applied to estimate the probability of positive screening for a current depressive episode (SCID-5) according to EPDS.” (Line 141 to 146).

# 3.7 Line 155, statistical software and the level of significance to be stated.

Our answer: The statistical software and significance level used was included as suggested: “The sample size was estimated to evaluate the accuracy of the screening scales for MD diagnosis, with an estimated sensitivity of 90.0% and a suggested sample size of 60 cases. All statistics were two sided, and a 0.05 significance level was used (Statistica 10.0 - Statsoft®).” (Line 147 to 150).

Remark 4#

Results # 4.1 Baseline characteristics of the subjects to be included and presented in table form.

 Our answer: Two sentences about the sample characteristics were included as well as the suggested table (table 2): “The sample consisted mostly of multigravidas (98.0%). Considering that 63.8% of the patients did not use contraception, about 37.5% had unplanned pregnancies. The most important risk factors for pregnancy were hypertension (22.9%), previous preterm labor (19.7%), complications at last delivery (18.0%), hypothyroidism (16.4%), and obesity (13.1%). Psychiatric disorders in the family were found in 32 cases (52.4%). In 26 cases, the informant indicated the family member, mostly the mother (43.2%) and uncles or aunts (30.8%). 21.3% had used illegal substances, 14.7% had attempted suicide, and 14.7% suffered from anxiety.” (Line 152 to 160).

# 4.2 Line 171, Graph 1 to be replaced with Figure 1.

Our answer: All graphical terms were replaced by the term figure as suggested.

# 4.3 Line 182, the cut off for Bipolar Disorder SCID-5 to be stated.

 Our answer: The diagnosis of bipolar disorder was stated using the SCID-5 interview (clinical version – mood disorders module) as the gold standard. The MDQ cut-off in this sample was greater than or equal to 7. A sentence with this description was included in the manuscript: “The cut-off point ≥ 7 of the MDQ showed an accuracy of 70.5% for identifying bipolar disorder, with a sensitivity of 70.0%, a specificity of 70.6%, a false-positive of 68.2%, and a false-negative of 7.7%”. (Line 181 to 183).

# 4.4 Table 2 Depressive Episode SCID-5, the percentage figures were presented based on row but for total, the percentage figures was based on column. This needs to be standardized.

Our answer: We replaced table 2 to Table 3 (adjusting it along the text), and the total percentages were removed. (See Table 3).

# 4.5 Graph 1, the dependent and independent variable to be clearly denoted in the footnote.

Our answer: The suggested information has been included in the Figure’s 1 legend.

Remark 5# Discussion

# 5.1 Line 255, the word 'chart 1' to be replaced with table and respective table number or to be omitted.

Our answer: All chart terms were replaced by the term table as well as its numbering.

# 5.2 Line 259 Chart 1, typo Sensibility.

 Our answer: It has been corrected. Thank you.

# 5.3 Decimal points for percentages to be standardized.

Our answer: It has been corrected. Thank you.

# 5.4 Not all references conform to PLoS One format.

 Our answer: The references were adjusted to Vancouver style as required by Plos One, thank you.

Reviewer #2: 

Remark 1# L70 "MDQ is a screening tool for episodes of mania" => The MDQ is a screening tool for history of mania or hypomania

Our answer: The sentence was corrected: “The MDQ is a screening tool to investigate history of mania or hypomania”. (Line 66-67).

Remark 2# L87 "This cross-sectional study included 61 childbearing women" The sample size is small and should be discussed.

 Our answer: The sample size was calculation indicated a minimum sample size of 60 subjects and this was further explained in the manuscript, as reviewer 1 also indicated: “The sample size was calculated to evaluate the accuracy of the screening scales for MD diagnosis, with an estimated sensitivity of 90.0% and the suggested sample size was 60 subjects. All statistics were two sided, and a 0.05 significance level was used (Statistica 10.0 - Statsoft®).” (Line 147 to 150).

Nevertheless, we attached possible ways for future studies with increased sample after the pandemic scenario that might be compared with this study and included a sentence in the limitation study: “Furthermore, given the prevalence of the bipolar disorder in this study, even eight times greater than in the general population, the MDQ results should be viewed with caution as a result of the sample size, although it was sufficient to indicate the significant contribution of the EPDS instrument to the diagnosis. Future studies with increased sample after the pandemic scenario using these screening tools at routine second-trimester examinations could help expand the sample to include childbearing women at all levels of care. They could also highlight the public benefits of perinatal mental health and demonstrate that health professionals can use simple tools to promote mental wellbeing”. (Line 319 to 326).

Remark 3# L89 High risk pregnancies group should be defined. Moreover, the reason why high risk pregnancies have been chosen to evaluate MDQ and EPDS effectiveness should be explained.

Our answer: This information was included in the manuscript: “This cross-sectional study included 61 childbearing women, all over 18 years, with gestational age between 18 and 24 weeks who attended prenatal consultations at a public university hospital reference for high-risk pregnancies in southern Brazil. These patients are referred by a Health Unit to the tertiary attention because they have or acquires a condition during pregnancy that puts them at higher risk for adverse events and requires more intensive care by obstetricians and other specialists [19]. The health care team must be prepared for many factors that may negatively impact high-risk pregnancies, whether they are clinical, obstetric, socioeconomic, or emotional [19]. According to Tsakiridis et al., prenatal depression is more likely in high-risk pregnancies because they have more risk factors, suggesting greater attention to screening these patients [20].” (Line 83 to 93).

Remark 4# L279 "Compared to the general female population, childbearing women have lower suicide rates." Nevertheless, suicide is one of the leading cause of maternal death in several countries.

Our answer: we appreciate the suggestion and modified the sentence: “Compared to the general female population, childbearing women have a lower rate of suicide; however, suicidal ideation may be more representative among the ones with depressive disorder [1,25]. Nevertheless, suicide is a leading cause of maternal death in several countries [29-32]. Therefore, it is essential to ask about suicidal thoughts in patients with depressive symptoms.” (Line 261 to 265).

Thank you again for the useful suggestions. We are willing to elucidate and improve any doubts or suggestions that may persist.

Regards,

Fernanda Schier de Fraga and Team.

---

## [Decision Letter · Decision Letter 1]

7 Dec 2021

PONE-D-21-26356R1Antenatal screening of depressive and manic symptoms in south Brazilian childbearing women: A transversal study in advance of the pandemic scenarioPLOS ONE

Dear Dr. Schier de Fraga,

Thank you for submitting your manuscript to PLOS ONE. After careful consideration, we feel that it has merit but does not fully meet PLOS ONE’s publication criteria as it currently stands. Therefore, we invite you to submit a revised version of the manuscript that addresses the points raised during the review process.

Please improve the manuscript according to Rev. 1 last comments.

We look forward to receiving your revised manuscript.

Kind regards,

Raoul Belzeaux, M.D.

Academic Editor

PLOS ONE

Journal Requirements:

Reviewers' comments:

Reviewer's Responses to Questions

**Comments to the Author**

1. If the authors have adequately addressed your comments raised in a previous round of review and you feel that this manuscript is now acceptable for publication, you may indicate that here to bypass the “Comments to the Author” section, enter your conflict of interest statement in the “Confidential to Editor” section, and submit your "Accept" recommendation.

Reviewer #1: All comments have been addressed

Reviewer #2: All comments have been addressed

2. Is the manuscript technically sound, and do the data support the conclusions?

Reviewer #1: (No Response)

Reviewer #2: Yes

3. Has the statistical analysis been performed appropriately and rigorously? 

Reviewer #1: (No Response)

Reviewer #2: I Don't Know

4. Have the authors made all data underlying the findings in their manuscript fully available?

Reviewer #1: (No Response)

Reviewer #2: Yes

5. Is the manuscript presented in an intelligible fashion and written in standard English?

Reviewer #1: (No Response)

Reviewer #2: Yes

6. Review Comments to the Author

Reviewer #1: Minor comments

Table 3, symbol n(%) to be added. For total, 100% to be added apart from total number.

Line 264, typo 'countries. [29-32] Therefore''

Reviewer #2: I have no futher comment about this study. All my comments about the previous round review were corrected.

7. PLOS authors have the option to publish the peer review history of their article (what does this mean?). If published, this will include your full peer review and any attached files.

Reviewer #1: No

Reviewer #2: No

---

## [Author Response · Author response to Decision Letter 1]

7 Dec 2021

Response to the reviewers

Dear Editor, 

Raoul Belzeaux 

Plos One

Thank you for the pertinent review and for the opportunity of improving our paper entitled “Antenatal screening of depressive and manic symptoms in south Brazilian childbearing women: A transversal study in advance of the pandemic scenario.”

 Please find below our answers. The changes are highlighted in yellow in the manuscript.

Reviewer#1

Remark1# Table 3, symbol n(%) to be added. For total, 100% to be added apart from total number.

Our answer: The suggestion has been included in Table 3. Thank you.

Remark 2# Line 264, typo 'countries. [29-32] Therefore''

 Our answer: It has been corrected. Thank you.

Reviewer #2: I have no futher comment about this study. All my comments about the previous round review were corrected.

Our answer: Thank you.

Thank you again for the useful suggestions. We are willing to elucidate and improve any doubts or suggestions that may persist.

Regards,

Fernanda Schier de Fraga and Team.

---

## [Editor Report · Decision Letter 2]

13 Dec 2021

Antenatal screening of depressive and manic symptoms in south Brazilian childbearing women: A transversal study in advance of the pandemic scenario

PONE-D-21-26356R2

Dear Dr. Schier de Fraga,

We’re pleased to inform you that your manuscript has been judged scientifically suitable for publication and will be formally accepted for publication once it meets all outstanding technical requirements.

Kind regards,

Raoul Belzeaux, M.D.

Academic Editor

PLOS ONE
---

## [Editor Report · Acceptance letter]

16 Dec 2021

PONE-D-21-26356R2 

Antenatal screening of depressive and manic symptoms in south Brazilian childbearing women: A transversal study in advance of the pandemic scenario 

Dear Dr. Schier de Fraga:

I'm pleased to inform you that your manuscript has been deemed suitable for publication in PLOS ONE. Congratulations! Your manuscript is now with our production department. 

Kind regards, 

on behalf of

Dr. Raoul Belzeaux 

Academic Editor

PLOS ONE